# A Lightweight High Definition Mapping Method Based on Multi-Source Data Fusion Perception

Haina Song [1] , Binjie Hu [1], Qinyan Huang [2], Yi Zhang [3] and Jiwei Song [4,*]

1   Department of School of Electronic and Information Engineering, South China University of Technology, Guangzhou 510641, China
2   Guangzhou Jiaoxintou Technology Co., Ltd., Guangzhou 510100, China
3   Guangzhou Laneposition Technology Co., Ltd., Guangzhou 511455, China
4   China Electronics Standardization Institute, Beijing 100007, China
*   Correspondence: songjw@cesi.cn

**Abstract:** In this paper, a lightweight, high-definition mapping method is proposed for autonomous driving to address the drawbacks of traditional mapping methods, such as high cost, low efficiency, and slow update frequency. The proposed method is based on multi-source data fusion perception and involves generating local semantic maps (LSMs) using multi-sensor fusion on a vehicle and uploading multiple LSMs of the same road section, obtained through crowdsourcing, to a cloud server. An improved, two-stage semantic alignment algorithm, based on the semantic generalized iterative closest point (GICP), was then used to optimize the multi-trajectories pose on the cloud. Finally, an improved density clustering algorithm was proposed to instantiate the aligned semantic elements and generate vector semantic maps to improve mapping efficiency. Experimental results demonstrated the accuracy of the proposed method, with a horizontal error within 20 cm, a vertical error within 50 cm, and an average map size of 40 Kb/Km. The proposed method meets the requirements of being high definition, low cost, lightweight, robust, and up-to-date for autonomous driving.

**Keywords:** crowdsourced data; multi-source data fusion perception; semantic alignment; semantic aggregation; high-definition map



## 1. Introduction

Currently, high-definition maps (HD maps) play a critical role in the accurate positioning of autonomous vehicles, whether it is for advanced driving assistance systems or full self-driving systems [1–5]. However, the traditional HD mapping scheme is costly and complex, resulting in low efficiency and update frequency levels, which hinders the widespread adoption of autonomous driving technology [6–8]. To address this challenge, researchers have turned to low-cost, fully automated, and crowdsourced mapping methods [9–15].

Crowdsourced mapping methods, based on low-cost sensors, are increasingly being explored to reduce mapping costs and promote the development of autonomous driving technology [14,15]. However, due to the large measurement and system errors of low-cost sensors, a semantic alignment fusion algorithm is necessary to optimize the data collected from multiple trajectories of the same road section to improve mapping accuracy [16–18].

Common registration algorithms include the iterative closest point (ICP) algorithm, the generalized iterative closest point (GICP) algorithm, and the semantic iterative closest point (SICP) algorithm [16–18]. While the ICP algorithm corrects the relative error between trajectories by constructing loop-closure pose constraints between multiple trajectories in a pairwise mode, it is heavily dependent on initial values and can easily be trapped in a local optimal solution, resulting in low registration accuracy. On the other hand, the GICP algorithm combines the ICP algorithm and the point-to-plane ICP into the probability frame model, eliminating the role of some bad corresponding points in the solution process.

Finally, the SICP algorithm improves on the ICP algorithm by adding semantic prior information, which results in higher alignment accuracy and faster convergence speed.

After the semantic alignment of multiple trajectories, a lightweight vector semantic map can be obtained by merging maps on the cloud server—a process known as semantic aggregation. Two different semantic aggregation methods have been proposed. Qin et al. [14] used rich, high-precision sensors to build local segment maps (LSMs) of the vehicle, and only semantic element aggregation was needed on the cloud. Their semantic aggregation method generates a grid map on the cloud and divides the LSM into 0.1 m × 0.1 m × 0.1 m small grids, each containing the location, the semantic element type, and the number of each semantic element type. The global map is merged in the cloud according to the grid of the same resolution, and the semantic element type with the highest score in each grid is used as the semantic type after aggregation. This method is more suitable for the local map generated by sensors with high accuracy and low noise tolerance. In contrast, Herb et al. [15] used crowdsourced data from commercial sensors and fused multi-period vehicle terminal maps in the cloud. The core of their alignment algorithm is to extract the landmarks observed in the keyframe of the sub-map common view, match the features, calculate the relative pose as a closed-loop constraint, and finally, construct the vector map through semantic map reconstruction. However, this method is computationally expensive and inefficient, and it requires a large storage bandwidth.

In summary, this paper proposes an improved semantic alignment algorithm and semantic clustering algorithm for lightweight mapping by merging LSMs on the cloud server. This low-cost, crowdsourced semantic mapping solution has the potential to reduce mapping costs and promote the development and application of autonomous driving technology.

## 2. Mapping System Overview

The HD mapping system is comprised of two components: real-time local mapping on the vehicle and global mapping on the cloud server, as depicted in Figure 1.

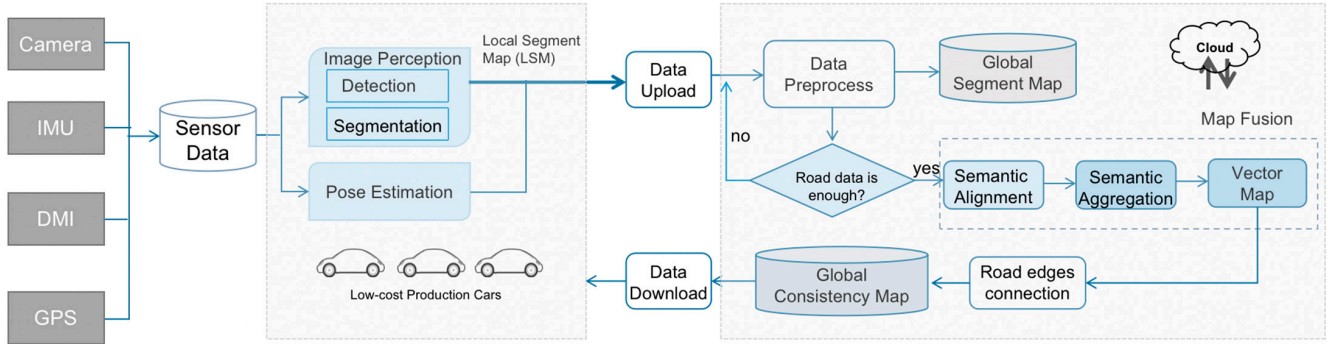

**Figure 1.** The architecture of HD mapping system.

Low-cost sensors, such as a monocular camera, an inertial measurement unit (IMU)/ global positioning system (GPS), and a wheel encoder, are used to collect data on-board the vehicle. Semantic elements, including lane lines, ground arrows, road stop lines, traffic signs, electrical poles, etc., are extracted from monocular vision. The ego-motion, estimated by the IMU and wheel encoder, is projected onto the 3D world coordinate system to form a local segment map (LSM) using semantic splicing. The global absolute pose of the vehicle can be estimated through GPS/IMU/wheel encoder fusion processing and then uploaded to the cloud server.

Limited by the precision of low-cost sensors and the computing power of the vehicle-end hardware platform, the LSMs generated through single-vehicle acquisition at one time cannot meet the requirements of semantic element precision and completeness for autonomous driving. To solve this problem, we can repeat LSM mapping for the same road

section many times and then upload LSMs to the cloud for overall fusion mapping, which can offset the observation error of single mapping, to some extent, and greatly improve the mapping precision and semantic completeness. First, the LSMs collected on the same road section are unified to the global world coordinate system. Then, semantic alignment is performed, different types of semantic elements are instantiated, and finally, the global semantic map is generated using vectorization.

However, the precision of low-cost sensors and the computing power of the vehicle-end hardware platform limit the LSMs' precision and completeness. To address this issue, LSM mapping is repeated multiple times for the same road section and then uploaded to the cloud server for overall fusion mapping. This offsets the observation error of single mapping to some extent and significantly improves the mapping precision and semantic completeness. The LSMs collected on the same road section are unified in the global world coordinate system, and semantic alignment is performed to instantiate different types of semantic elements. Finally, the global semantic map is generated through vectorization.

Due to the large measurement error and system error of low-cost sensors, we propose a fusion alignment algorithm, GBA_SGICP, based on crowdsourced data, for semantics alignment. we also present an improved density clustering algorithm for semantic aggregation in this paper.

## 3. Local Mapping on Vehicle

### 3.1. Semantic Segmentation and Detection

To improve the accuracy and efficiency of image perception, semantic segmentation and object detection methods were used to extract semantic features from images [19–25]. To further advance this approach, a neural network model, called OneNet, was proposed. OneNet integrates the detection and segmentation of all semantic elements and is composed of an encoding–decoding main network, target detection head, semantic segmentation head, and instance segmentation head. Figure 2 shows the architecture of OneNet.

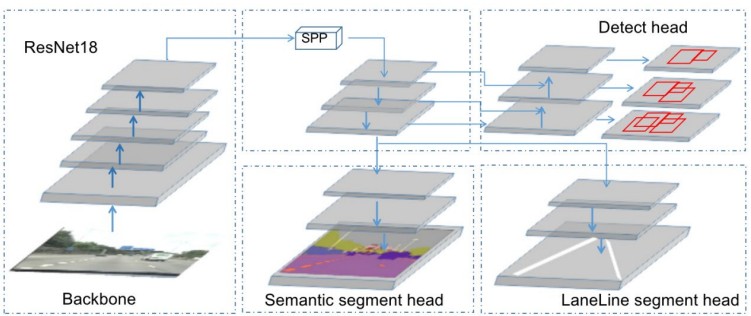

**Figure 2.** The architecture of OneNet.

In OneNet, ResNet18 is adopted as the backbone network to extract features from input images [26]. The target detection head is used to extract air elements, such as traffic lights, traffic signs, and other elements, and the semantic segmentation head is responsible for extracting ground elements, such as arrows, dotted line boxes, lane stop lines, etc. [27]. The instance segmentation head is used to extract lane line semantic elements [28]. The effectiveness of this approach is demonstrated in Figure 3, which shows the segmentation detection results [29].

### 3.2. Semantic Element Ranging

To obtain the local map under 3D world coordinates, the semantic elements extracted from the OneNet model are projected into the vehicle body coordinate system. Since monocular vision is used for local mapping, distance measurement is necessary to estimate the depth of semantic pixels. For the ground elements, the pixel depth is estimated primarily based on the assumption of a near-ground plane. To improve ranging accuracy, a ground equation calibration algorithm, based on the vanishing points of lane lines, is used [30]. For

aerial elements, BA optimization is performed through optical flow tracking and attitude, provided by ego-motion, to estimate the image depth [31].

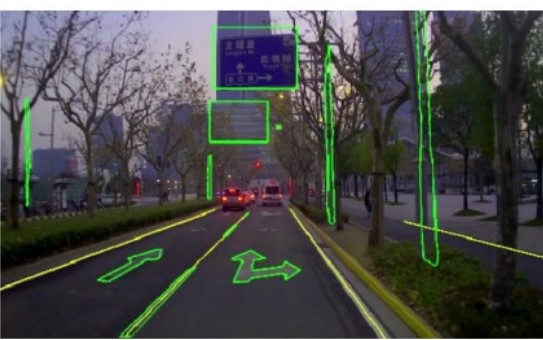

**Figure 3.** Example of semantic segmentation and detection results of local mapping on a vehicle.

### 3.2.1. Near-Ground Plane Assumption

It is assumed that the ground can be approximated as a plane within a certain range of camera observations. This allows for a fixed-projection relationship between 2D points on the camera image plane and 3D points on the ground, which can be directly back-projected to solve the problem of monocular camera ranging for ground elements [32]. By estimating the camera pose and using the ground equation calibration algorithm based on the vanishing points of lane lines, the ranging accuracy of the ground elements can be improved.

### 3.2.2. Ground Element Ranging

To estimate the depth of ground elements, the near-ground plane assumption is commonly used. The derivation formula is as follows:

$$X_t = \frac{-hK^{-1}x_t}{n^T K^{-1}x_t} \tag{1}$$

where $X_t$ represents the feature point on a 2D image captured by the monocular camera at time $t$, which corresponds to pavement features, such as lane lines and ground arrows. $K$ is the camera's internal parameter, which is a $3 \times 3$ linear projection matrix. By inverting $K$, we can obtain the points on the normalized projection plane using $X_t$. The normal vector of the ground plane equation is denoted by $n$, and $h$ represents the camera height above the ground, both of which can be obtained through external parameter calibration. Using these parameters, we can recover the depth information of the 3D points corresponding to the ground features under the assumption of a near-ground plane.

### 3.2.3. Aerial Element Ranging

The ranging of aerial features is accomplished through target tracking and geo-motion triangulation. Target tracking is based on perceptual semantic detection instance, meaning that only aerial semantic elements detected through perception are tracked. In monocular vision, triangulation introduces scale uncertainty. To address this, the algorithm leverages ego-motion to obtain the relative pose between two tracking frames, which is equivalent to obtaining the baseline of the next two observations (i.e., binocular vision). This approach resolves the scale problem of monocular ranging.

The derivation formula is:

$$p_i^w = argmin \sum_{j=1}^{M} \|r_{ij}\|^2, \tag{2}$$

$$r_{ij} = z_{ij} - \pi\left(p_i^w, R_w^{c^j}, t_{c^j}^w\right) \tag{3}$$

where $\left\{ R_w^{cj}, t_{cj}^w \right\}_{j=1,\dots,M}$ is the pose of the monocular camera at different times, which can be obtained from the ego-motion calculation, and $p_i^w$ denotes the spatial position of the 3D feature points of semantic instance $i$ in the world coordinate system. Examples here include, but are not limited to, traffic signs, lights, poles, etc. Semantic segmentation and detection are implemented on the same network. $z_{ij}$ denotes tracked 2D feature points, and the optical flow method is used to track and determine multi-frame 2D semantic observations belonging to the same aerial landmark, where $j$ is used to identify multi-frame observations belonging to the same semantic instance, $i$. $\pi(.)$ is a reprojection function, and the Gauss–Newton method or the LM method is usually used to minimize the reprojection error to find the optimal $p_i^w$.

After extracting the 3D semantic features from consecutive multi-frame images, they are transformed into the world coordinate system, with the first frame as the origin using the ego-motion information, $n$. Then, the transformed multi-frame semantic elements are merged to generate a local semantic map [33]. An example of the local semantic map is illustrated in Figure 4.

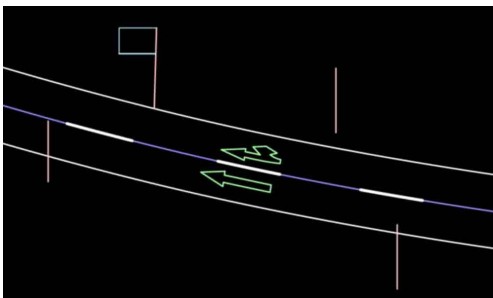

**Figure 4.** Example of structural semantic element map.

## 4. Fusion Mapping on Cloud Server

In this section, we will discuss the steps and improved algorithm of fusion mapping on-cloud. The following are the details of the method:

(1) First, all local semantic maps (LSMs) generated by the vehicle are transferred from the local coordinate system to the global coordinate system through global pose graph optimization, which ensures that all LSMs are aligned to the same global reference frame.

(2) Next, semantic alignment or trajectory alignment is performed to improve the position accuracy of semantic elements in the global coordinate system by optimizing the poses of multiple trajectories. This step is crucial for determining the quality of the resulting map.

(3) Finally, semantic elements are aggregated to instantiate aligned semantic elements and generate vector semantic maps.

The structure of on-cloud fusion mapping is illustrated in Figure 5.

### 4.1. Global Pose Graph Optimization

In order to build a globally consistent semantic map, it is necessary to unify the local trajectory and semantic elements of LSMs for different vehicles and time periods with the global coordinate system (ENU coordinate system) since they are relative to the vehicle body coordinate system at a certain time. This is achieved by aligning the odometry pose of the LSMs with the global coordinate system. To accomplish this, we construct global pose graph optimization by fusing the LSM-associated local trajectory and GPS trajectory, as shown in Figure 6a. The optimization problem can be expressed as:

$$\chi^* = \arg\max_{\chi} \sum_{t=0}^{n} \left\| e^l(z_{t-1,t}^l, \chi) \right\|_{\Omega_t^l}^2 + \left\| e^g(z_t^g, \chi) \right\|_{\Omega_t^g}^2 \tag{4}$$

$$e^l(z^l_{t-1,t}, \chi) = \begin{bmatrix} q^l_{t-1}{}^{-1}(p^l_t - p^l_{t-1}) \\ q^l_{t-1}{}^{-1}q^l_t \end{bmatrix} \ominus \begin{bmatrix} q^w_{t-1}{}^{-1}(p^w_t - p^w_{t-1}) \\ q^w_{t-1}{}^{-1}q^w_t \end{bmatrix} \tag{5}$$

$$e^g(z^g_t, \chi) = p^{GPS}_t - p^w_t \tag{6}$$

where $\Omega^l_t$ and $\Omega^g_t$ denote the covariance of odometer error and GPS measurement error, respectively, $(p^l_{t-1}, q^l_{t-1})$ represents the local pose at $t-1$, and $e^l$ and $e^g$ represent the relative pose constraint and the global absolute position constraint, respectively. The optimization goal is $\chi = \{x_0, x_1, \ldots, x_n\}$, where $x_i = \{p^w_i, q^w_i\}$.

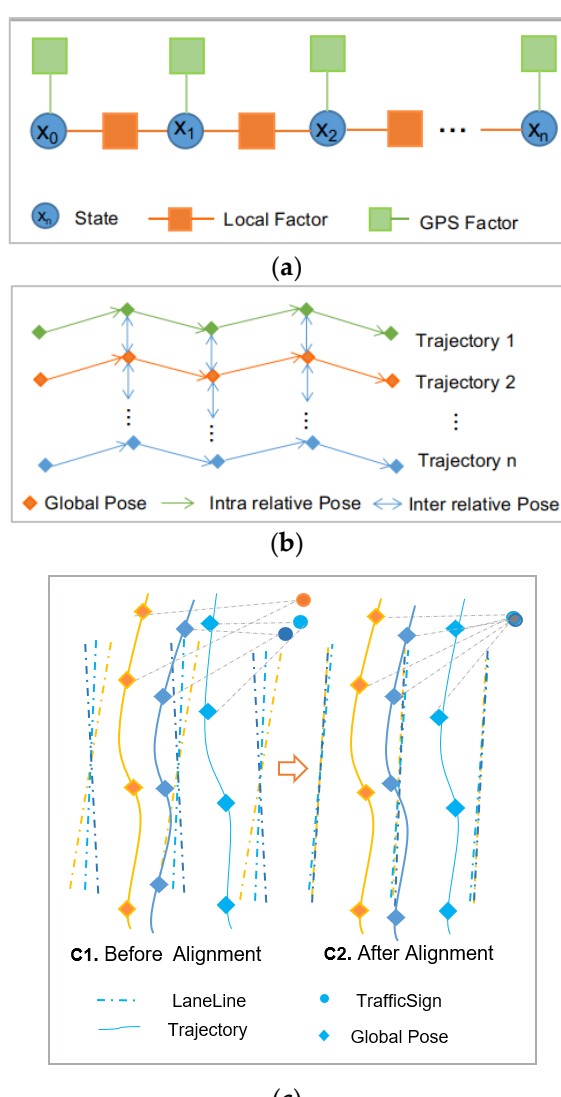

**Figure 5.** The structure of the fusion mapping on-cloud algorithm. (**a**) Global pose graph optimization; (**b**) semantic alignment; (**c**) semantic aggregation. Different color line represents different trajectory.

### 4.2. Semantic Alignment Algorithm

Due to the low measurement accuracy of low-cost sensors, even though the LSM data constructed using different trajectories are converted to the unified global coordinate system in the cloud, there is still a certain pose deviation, as shown in Figure 5C-c1. Therefore, it is necessary to use a semantics alignment fusion algorithm to correct the relative relationship between trajectories. The semantic alignment algorithm is the core of on-cloud fusion mapping using crowdsourced data, which will directly determine the accuracy of fusion mapping.

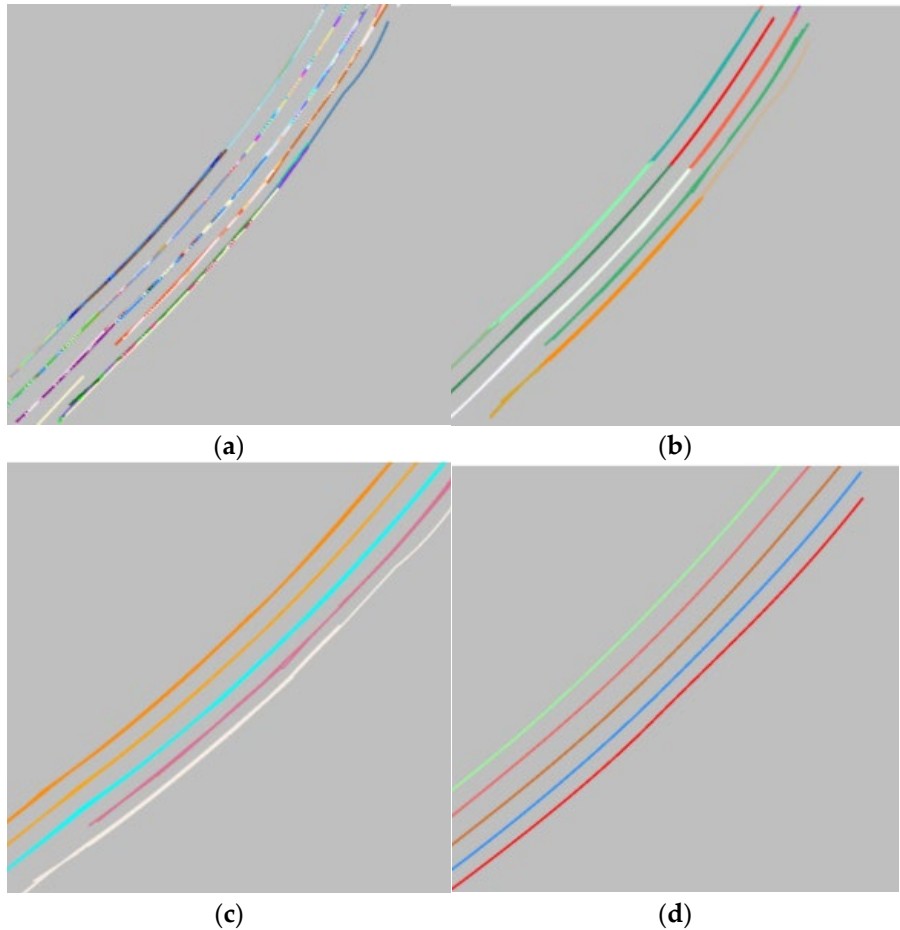

**Figure 6.** Schematic diagram of lane aggregation algorithm. (**a**) LSM lane lines, where different colors represent different lane lines; (**b**) effect picture after the first stage of clustering, where different colors represent different clusters; (**c**) effect picture after clustering in the second stage, where the same color represents one instance; and (**d**) results after vectorization.

To address this issue, we propose a two-stage, multi-trajectory pose optimization algorithm, called global bundle adjustment, based on semantic GICP (GBA_SGICP), which is based on SICP and GICP improvement.

In the first stage, we add semantic prior information to GICP, use different loss functions for different element types, such as ground arrows and traffic signs, and build a loss function based on the point-to-plane distance to reduce the point-cloud matching error. To make the registration position converge to a more accurate value, we use a semantic pyramid hierarchy, where the upper semantic registration result serves as the initial value of the lower registration. We refer to this improved algorithm as the semantic GICP (SGICP).

Next, we use the SGICP algorithm to calculate the inter-relative pose constraint (InterRPC) between trajectories using the pairwise method. We then add the relative constraints in each trajectory to construct a graph optimization to obtain a more accurate pose, which improves the robustness of the algorithm in different scenarios and enhances the convergence speed of the algorithm [34,35]. Figure 3 shows that the semantic alignment fusion algorithm corrects the relative relationship between trajectories, which is essential for achieving accurate fusion mapping.

The SGICP algorithm solves the problem as follows:

Assuming $P_s$ is the source point cloud (point cloud to be registered), $P_t$ is the target point cloud, $P_s^k$ and $P_t^k$ represent the source and target point clouds of the $k$-th layer of the pyramid, respectively, and $T \in SE(3)$ is the required rotation translation matrix. Then,

we have $f_{SGICP}(T) = \sum_{l=1}^{L} r(T|P_s^k, P_t^k, \mathcal{A}(l))$, $P_s^k = \frac{1}{2^{k-1}} P_s$, $P_t^k = \frac{1}{2^{k-1}} P_t$, where $\mathcal{A}(l)$ is the matching relationship of $P_s^k$ and $P_t^k$, and the $K$ nearest neighbor algorithm is usually used.

$$r(T|P_s^k, P_t^k, \mathcal{A}(l)) = \sum_{i=1}^{n} w_l \| P_t^{ki} - T * P_s^{ki} \|_{C_i}^2, \ C_i = C_i^t - TC_i^s T^T \tag{7}$$

where $i$ is the number of semantic point cloud pairs matched by $\mathcal{A}(l)$, $w_l$ denotes the confidence weight of different semantic elements, $P_s^{ki}, P_t^{ki}$ represents the $i$-th matching point of the $k$-th layer source point cloud and target point cloud of the pyramid respectively. $\{C_i^s\}$ and $\{C_i^t\}$ denotes the covariance matrix of source point cloud and target point cloud measurements at the $i$ th point. Optimization variables $T^* = arg\min_{T \in SE(3)} f_{SGICP}(T)$.

Semantic GICP algorithm builds the loop-closure pose constraint between multiple dependent trajectories, that is, using $C = \left\{ c_{i_m}^{j_n} \right\}$ to represent the observation value of loop-closure constraint between multiple trajectories, $i, j \in \{1, 2, \cdots, R\}$, $i \neq j$.where $R$ is the number of all trajectories, and $c_{i_m}^{j_n}$ represents the loopback constraint between the m-th pose node of the $i$-th trajectory and the n-th pose node of the $j$-th trajectory.

InterRPC builds constraints between different trajectories. When the relative constraint between adjacent pose nodes within each trajectory is considered at the same time, each pose node of multiple trajectories can be constructed $X = \left\{ \chi_i^r \right\}_{i=1,\cdots,N^r}^{r=1,\cdots,R}$, $\chi_i^r$ represents the ith pose node of the r-th trajectory, $\chi = (R, t) \in SE(3) \times \mathbb{R}^{4 \times 4}$. Let the observation value of adjacent pose constraints and prior constraints in each trajectory $Z = \left\{ u_i^r \right\}_{i=1,\cdots,N^r}^{r=1,\cdots,R} \cup \left\{ p_i^r \right\}_{i=1,\cdots,N^r}^{r=1,\cdots,R}$.

The optimization problem can be described using probabilistic model as:

$$P(X|Z, C) \propto \prod_{r=1}^{R} \left( \prod_{i=1}^{N^r} P(\chi_i^r|p_i^r) \prod_{i=1}^{N^r} P(\chi_{i+1}^r|\chi_i^r, u_i^r) \right) \prod_{ij}^{R} P\left( \chi_i^m|\chi_j^n, c_{i_m}^{j_n} \right) \tag{8}$$

We assume that the measurement model follows the Gaussian distribution, and the relative pose constraint probability model in the trajectory is given by

$$P(\chi_{i+1}^r|\chi_i^r) = N(f_i(\chi_i^r, u_i^r), \Lambda_i^r) \tag{9}$$

The relative pose constraint probability model between different trajectories is given by

$$P(\chi_i^m|\chi_j^n) = N\left( h_i\left( \chi_j^n, c_{i_m}^{j_n} \right), \Gamma_{i_m}^{j_n} \right) \tag{10}$$

$X^*$ under the maximum a posteriori probability is the optimal solution we obtained. You can take the negative logarithm of Formula (8) and combine Formula (9) and (10) to convert it into a nonlinear least squares problem

$$\begin{aligned}
X^* = \underset{X}{argmax} P(X|Z, C) &= \underset{X}{argmin} - logP(X|Z, C) \\
&= \underset{X}{argmin} \left\{ \sum_{r=0}^{R} \left( \sum_{i=1}^{N^r} \| p_i^r - \chi_i^r \|_{\Sigma}^2 + \sum_{i=1}^{N^r} \| f_i(\chi_i^r, u_i^r) - \chi_{i+1}^r \|_{\Lambda_i^r}^2 \right) + \sum_{ij}^{M} \rho \left( \| h_i\left( \chi_j^n, c_{i_m}^{j_n} \right) - \chi_i^m \|_{\Gamma_{i_m}^{j_n}}^2 \right) \right\}
\end{aligned} \tag{11}$$

Considering the robustness of the algorithm, we add a filter function $\rho(.)$ of the pose edge in loop-closure pose in the Formula (11) Pairwise Consistency Measurement (PCM) [30] based on maximum clique is used to filter out abnormal or low precision loopback pose edges.

The optimized pose can be obtained The optimized pose $\chi = (R, t)$, by solving the aforementioned nonlinear least square problem. The position error of semantic elements caused by trajectory errors can be corrected by using the formula $P' = RP + t$ thus

achieving alignment of semantic elements observed by multiple trajectories. To perform iterative optimization, we use the Levenberg Marquard algorithm [36]. The algorithm is implemented using the C++ three-way toolkit GTSAM [37].

In the second stage, we use the pose calculated in the first stage as the initial value, and add the spatial constraints of semantic elements to construct a global Boundary Adjustment optimization. This method, called GBA_SGICP (Global Bundle Adjustment based on Semantic GICP), optimize both the relative error between multiple trajectories and the overall absolute error. This approach further improves the semantic alignment accuracy, as shown in Figure 6b. Unlike the traditional 2D-3D Boundary Adjustment [38], we use Direct Bundle Adjustment [39] to optimize the spatial position of corresponding corresponding 3D-3D semantic elements directly. For the semantic elements of line, such as lane line and stop line, the error function $e_l$ is based on the distance between the point and the line, expressed as:

$$e_l(x_s, x_t) = \sum_{k=1}^{N_L} \| \left( I - n_l^k n_l^{kT} \right) \left( x_s p_s^k - x_t p_t^k \right) \|_2^2 \tag{12}$$

where $x_s$ and $x_t$ denotes the pose node on the trajectories $s$ and $t$ respectively, $p_s^k$ and $p_t^k$ denotes the $k$-th semantic element matched pair observed at the pose nodes $x_s$ and $x_t$; $N_L$ is the set of all semantic features belonging to a point-line matching relationship; $n_l^k$ denotes the unit normal vector of the line where the position $p_t^k$ is located.

Similar to the semantic elements of a plane, such as ground arrows and traffic signs, the loss function, $e_p$, based on the distance between the point and the plane can be defined by

$$e_p(x_s, x_t) = \sum_{k=1}^{N_P} \| n_p^{kT} \left( x_s p_s^k - x_t p_t^k \right) \|_2^2 \tag{13}$$

where $N_P$ is the set of all semantic elements belonging to the point–plane matching relationship and $n_p^k$ denotes the unit normal vector of the plane where the position $p_t^k$ is located.

Let $\chi = \{x_1, x_2, \cdots, x_n\}$ be the set of all pose nodes of multiple trajectories, and $x_s, x_t \in \chi$. All matching pairs and initial pose values in (12) and (13) are derived from the optimization in the first step. The trajectory pose optimization equation in the second step can be obtained by using (12) and (13)

$$\chi^* = \underset{\chi}{argmin} \sum_{m,n}^{R} \sum_{i \in m, j \in n} \left[ \rho \left( e_l(x_i, x_j) \right) + \rho \left( e_l(x_i, x_j) \right) \right] \tag{14}$$

where $R$ denotes the number of tracks participating in fusion optimization, $m, n \in R, m \neq n$ represents the $m$-th and $n$-th tracks, $x_i, x_j$ denotes the $i$-th pose node of the $m$-th pass and the $j$-th pose node of the $n$-th pass, respectively. The Huber loss function, $\rho$, can be defined

$$\rho(s) = \begin{cases} s, \ s \leq 1 \\ 2\sqrt{s} - 1, \ s > 1 \end{cases}.$$

By employing the aforementioned optimization, we can obtain more precise trajectory poses, resulting in the improved alignment of semantic elements observed by multiple trajectories, as depicted in Figure 5c–c2.

### 4.3. Semantic Aggregation

After performing semantic alignment, the semantic elements observed by multiple tracks achieve good consistency in the global coordinate system, as shown in Figure 4A. However, this is still a composition of many semantic elements. To obtain a lightweight vector semantic map, it is necessary to aggregate the fusion map after multiple semantic alignments. The process of semantic aggregation includes instantiation and vectorization.

### 4.3.1. Instantiation

In this section, we propose a lane instantiation algorithm that is based on clustering and uses logical constraints to address the complexity of lane instantiation. The algorithm consists of a two-stage DBSCAN (density-based spatial clustering of applications with noise) [39] method.

In the first stage, we expand the LSM lane elements from the vehicle left and right within a certain range (20 cm) to form buffered line objects, which we denote as B. The input dataset is denoted as $D = \{b_1, b_2, \cdots, b_n\}$, where each b is a sample. We use the overlap area between two samples of D as the distance measurement of DBSCAN, expressed as $\mathbf{distance}(b_i, b_j) = \mathbf{overlap\_area}(b_i, b_j)$. The resulting clusters are marked as $C = \{c_1, c_2, \cdots, c_n\}$, as shown in Figure 6b.

The LSM lane lines with noise can be detected and eliminated using the first stage of clustering. Each cluster generated in the first stage of clustering forms a MultiLine object. In the second stage of clustering, we use the cluster set formed in the first stage as input, and we define the distance measurement of clustering as the Euclidean distance between two MultiLine objects. Considering the logical relationship between lane lines in the real world, a lane line can only be connected to, at most, the following lane lines, such as a new lane instance that is generated at the bifurcation.

We then adjust the definition of the core object in the DBSCAN algorithm. If the number of samples in the fruit subsample set meets the condition $\mathbf{MinPts}|N_\epsilon(c_j)| \leq \mathbf{MaxPts(MinPts = 1, \ MaxPts = 2)}$, then we add sample $c_j$ to the core object set and define the cluster generated by the second stage of clustering as $C' = \left\{c'_1, c'_2, \cdots, c'_n\right\}$, which is the final instantiation object of the lane, as shown in Figure 4C. This method can deal with any road scene, including branching roads, and has stronger robustness compared to other clustering methods.

### 4.3.2. Vectorization

Vectorization is the process of converting the same instance formed after clustering into a unique vector semantic element. For the lane line, we use third-order Bessel curve fitting to vectorize, which generates a smooth curve that closely approximates the lane line. The third-order Bézier curve can be expressed as:

$$B(t) = (1-t)^3 P_0 + 3(1-t)^2 t P_1 + 3(1-t)t^2 P_2 + t^3 P_3, \ 0 \leq t \leq 1 \tag{15}$$

where $P_0, P_1, P_2, P_3$ are the curve control points that define the shape of the curve and $t$ is a parameter that varies between $0$ and $1$. We select four points on the lane line instance as the control points and use the least-squares method to optimize their positions to fit the Bézier curve. The resulting curve is then used to represent the lane line in the vector semantic map. The vectorization result is shown in Figure 6d.

To instantiate discrete semantic elements, such as ground arrows, poles, traffic signs, etc., we define the distance measurement method of density clustering based on their spatial characteristics. For ground arrows, we calculate the intersection area between two samples, which is the overlapping area between two rectangular boxes. The clustering process uses the weighted average method for vectorization. To calculate the spatial relationship in all clustering processes, we use the C++ Boost.Geometry library.

## 5. Testing and Result Analysis

We conducted experiments, using crowdsourced data collected in real-world scenarios, to evaluate the effectiveness of our proposed GBA_SGICP algorithm and the capacity of fusion mapping on a cloud server. For this experiment, we used three vehicles equipped with the same sensors, including GPS, IMU, wheel speed encoder, and monocular camera, to collect data. The test scenario was a typical complex overpass scenario, as shown in Figure 7.

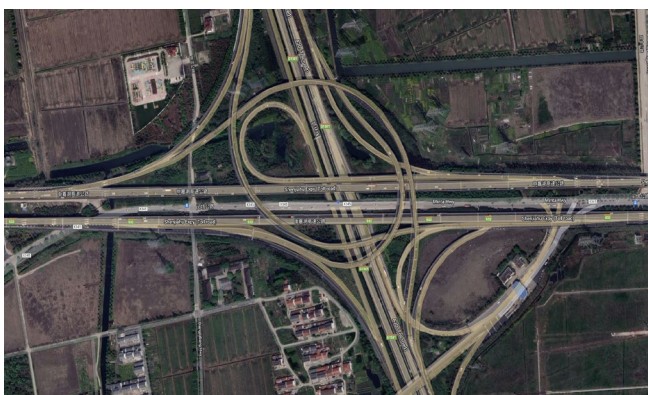

**Figure 7.** Test scenario.

To evaluate the quality of our automated crowdsourced mapping approach, we used the map produced by high-precision equipment acquisition vehicles and manually verified it as the ground truth. We used the following evaluation metrics: relative precision, semantic completeness, and semantic redundancy.

a.  Relative precision measures the accuracy of the relative position relationship between map elements, such as the lane width, the distance between two adjacent poles, etc. It is an important indicator to measure whether the map accuracy meets the requirements of autonomous driving. In this paper, we define relative precision as a comprehensive index of horizontal and vertical precision. The formula for calculating the relative precision of the same type of semantic elements is as follows:

$$\text{relative precision} = \frac{\sum_{i=1}^{N} \left| rd_i^{obs} - rd_i^{truth} \right|}{N}$$

where $N$ is the total number of semantic elements of the same type in the crowd-sourced map, $rd_i^{obs}$ is the relative distance of $i$-th elements of the same type in the crowdsourced map, and $rd_i^{truth}$ is the true value of the corresponding point.

b.  Semantic element completeness rate: It measures the proportion of map elements (lane line, traffic sign, traffic light, etc.) in the real world. It is also an important indicator of the automatic driving positioning system. The formula for calculating the completeness rate of the same type of semantic elements is

$$\text{semantic element completeness rate} = \frac{\sum_{i=1}^{N} Num_i^{obs} \left( \sum_{i=1}^{N} Len_i^{obs} \right)}{N \left( \sum_{i=1}^{N} Len_i^{truth} \right)}$$

c.  Semantic element redundancy rate: It refers to the proportion of elements constructed from maps that do not exist in the real world. It measures the degree of incorrect construction of map elements. The formula for calculating the redundancy rate of the same type of semantic elements can be expressed as:

$$\text{semantic element redundancy rate} = \frac{Sum\_Num^{obs} \left( Sum\_Len^{obs} \right)}{N \left( \sum_{i=1}^{N} Len_i^{truth} \right)}$$

For discrete-type elements (such as signboards), the calculation formula is $Sum\_Num^{obs}/N$; for continuous type elements (such as lane lines), the calculation formula is $Sum\_Len^{obs}/\sum_{i=1}^{N} Len_i^{truth}$, where $N$ is the number of instances of semantic elements of the same type that actually exist in the real world, and $Sum\_Num^{obs}/Sum\_Len^{obs}$ represents the number of elements that do not exist in the real world but exist in the map construction.

We conducted a comparison between our proposed GBA_SGICP algorithm and different alignment algorithms using the same dataset, and with a data concentration of 50 times

for each road. We evaluated the mapping accuracy of four alignment algorithms on four semantic elements, including two ground and two air elements, as shown in Table 1. The results indicate that our proposed GBA_ SGICP algorithm achieves the highest relative accuracy compared to the other alignment algorithms.

**Table 1.** Impact of cloud alignment algorithm on mapping accuracy (3-sigma avg.m).

| Method | LaneLine | LaneMarking | TrafficSign | Pole |
|---|---|---|---|---|
| ICP | 0.45 | 2.25 | 2.51 | 2.18 |
| SICP | 0.23 | 1.24 | 1.76 | 1.35 |
| SGICP | 0.19 | 0.72 | 0.83 | 0.66 |
| GBA_SGICP | 0.17 | 0.49 | 0.58 | 0.51 |

Table 2 presents the impact of the four alignment algorithms on the mapping efficiency using the average values of the efficiency of various scenes, including high-speed and urban scenes. The results show that the GBA_SGICP algorithm can achieve a mapping efficiency of nearly 18 km/h, which is about three times higher than the ICP alignment algorithm.

**Table 2.** Impact of cloud alignment algorithm on mapping efficiency (unit: km/h).

| Method | ICP | SICP | SGICP | GBA_SGICP |
|---|---|---|---|---|
| Efficiency | 6.4 | 10.2 | 13.5 | 17.7 |

In addition, we compare the low-cost lightweight crowdsourcing mapping method proposed in this paper with the two representative crowdsourcing cloud mapping methods in previous studies, such as the methods proposed by Qin et al. [14] and Herb et al. [15], respectively. The comparison of the accuracy, completeness, and overall map construction efficiency of the three fusion mapping algorithms for different semantic elements is shown in Table 3. It is shown that our method has the lowest accuracy error, higher factor completion rate, and relatively high efficiency.

**Table 3.** Performance comparison of three crowdsourced mapping methods.

| Object | LaneLine | | LaneMarking | | TrafficSign | | Pole | | Efficiency |
|---|---|---|---|---|---|---|---|---|---|
| Algorithm | Acc | Compl | Acc | Compl | Acc | Compl | Acc | Compl | |
| T Qin's method | 0.48 | 0.88 | 2.34 | 0.94 | 2.22 | - | - | - | 19.5 |
| M Herb's method | 0.20 | 0.93 | 1.21 | 0.85 | 1.58 | - | 1.28 | - | 3.8 |
| Our method | 0.17 | 0.96 | 0.49 | 0.95 | 0.58 | 0.89 | 0.51 | 0.91 | 17.7 |

We analyzed the completeness rate and redundancy rate of different semantic elements and found that the ground elements, such as lane line, ground arrow LaneMarking, and StopLine, were slightly better than air elements, such as TrafficSign and Pole, as shown in Figure 8. This is because the quality of ground elements in the LSM generated on-vehicle is better than that of air elements.

The comparative analysis of the influence of different observation times on fusion mapping on-cloud is shown in Figure 9. The results indicate that the accuracy of map elements improves as the observation time increases. For instance, the relative error of 50 observation fusions is over three times lower than that of 10 fusion mapping, and the element completeness rate is over double. When 100 fusions are reached, the relative accuracy of the lane line is close to 10 cm, and that of other elements is within 50 cm. Additionally, the completeness rate of elements is 95% on average, which has reached the mapping level of traditional high-precision sensors and can meet the needs of high-level

autonomous driving. Therefore, it can be concluded that increasing the observation time can significantly improve the accuracy and completeness of fusion mapping.

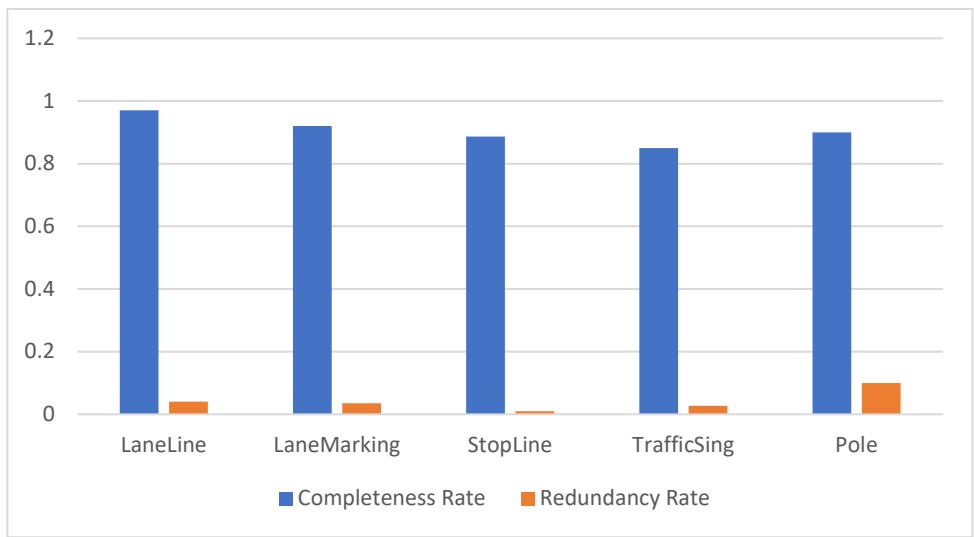

**Figure 8.** Statistical results of completeness and redundancy of crowdsourced fusion mapping on-cloud on different semantic elements.

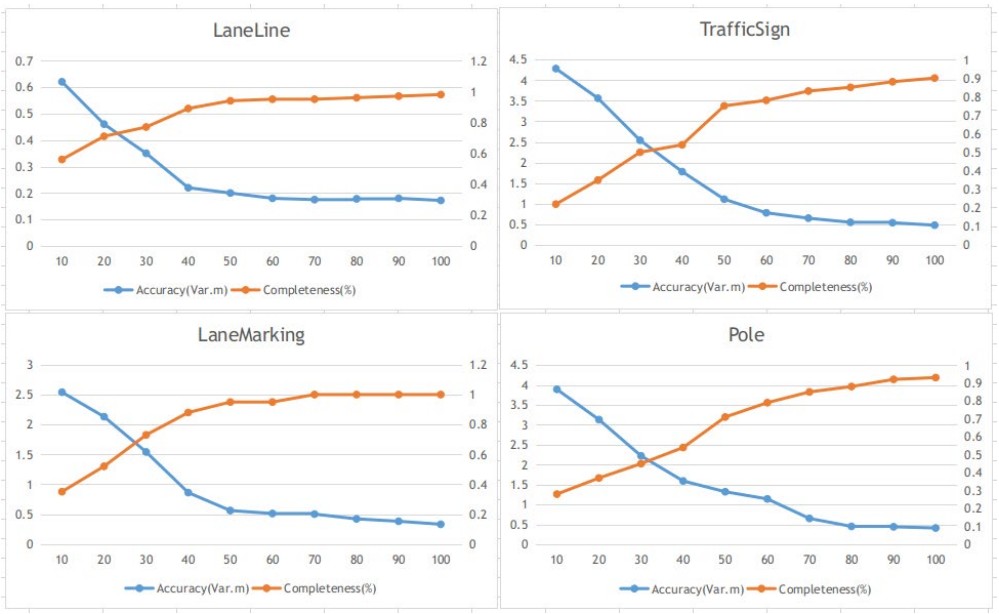

**Figure 9.** The influence of different observation times on the accuracy and completeness of fusion mapping, the horizontal axis represents the observation times (aggregation times), the left vertical axis represents the relative precision (unit: m), and the right vertical axis represents the semantic element completion rate.

We conducted an analysis of crowdsourced fusion mapping on a section of an urban road in an actual scene, and the results are presented in Figure 10. It is observed that due to the accuracy limitations of low-cost sensors, even though LMS data has been transferred to the global coordinate system through pose graph optimization, large errors still exist in multiple observation results of the same road section as shown in Figure 10a. However, after the optimization of semantic alignment, the location consistency of semantic elements observed many times has been greatly improved as shown in Figure 10b. Further, after semantic aggregation algorithm processing, some abnormal or low-precision observations

are filtered out, thus obtaining a high-precision lightweight semantic map as shown in Figure 10c.

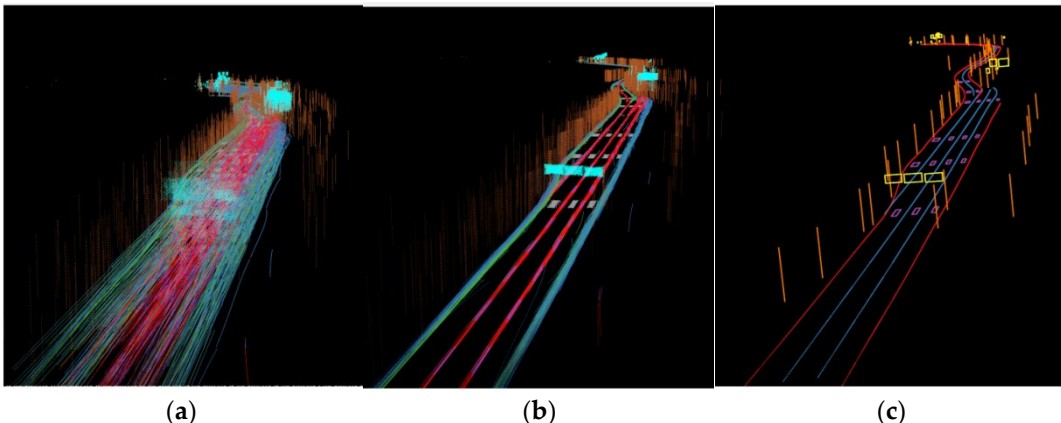

(**a**)    (**b**)    (**c**)

**Figure 10.** Effect of fusion mapping on-cloud process. (**a**) The original fragment map with multiple LSMs unified to the global coordinate system; (**b**) multiple fragment maps via GBA_SGICP algorithm optimization and semantic location adjustment; and (**c**) fragment map after aggregation and vectorization of semantic elements.

In this experiment, we utilized our crowdsourced semantic mapping method to automatically build a semantic map covering approximately 25 km in a complex circular overpass scenario, as shown in Figure 11. The semantic elements include various types of lanes (real lane lines, virtual lane lines, road edges, etc.), ground arrows, traffic signs, etc. To verify the actual usability of this map, we loaded the map onto a vehicle and conducted semantic positioning tests. The results show an average lateral error of 15 cm, a longitudinal error of 35 cm, and a heading angle error of 0.5 degrees, as shown in Figure 12. These results confirm the reliability and practicality of the crowdsourced mapping method proposed in this paper.

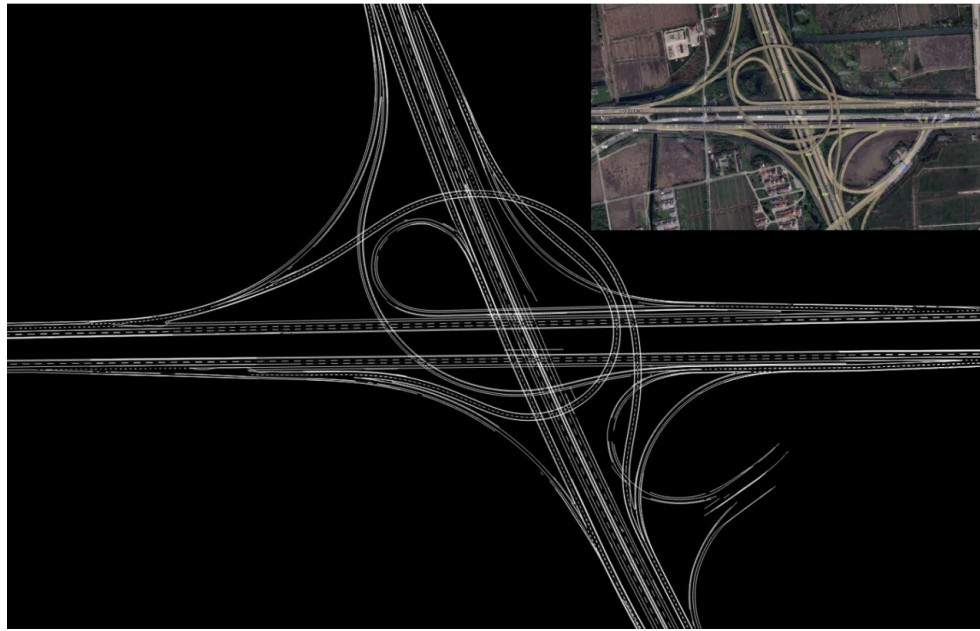

**Figure 11.** Automatic global consistency semantic map rendering.

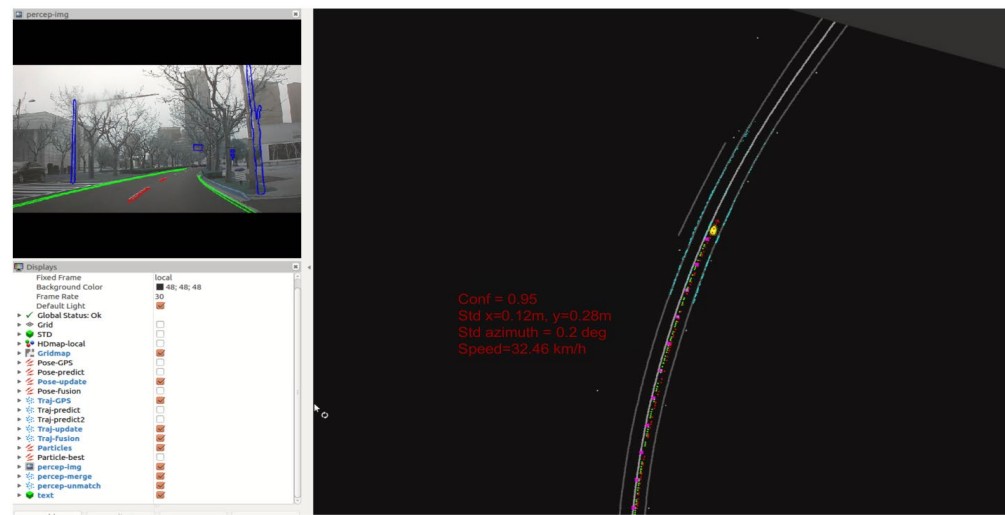

**Figure 12.** Actual effect picture of fusion positioning based on the crowdsourced map.

## 6. Conclusions

This paper proposes a low-cost and lightweight semantic HD mapping method for autonomous driving applications that is based on multi-source data fusion perception. The mapping system framework builds upon previous work that utilizes crowdsourced data, but the paper introduces an improved semantic alignment algorithm and semantic aggregation algorithm that are evaluated in a practical scene. For fusion mapping on the cloud, we focus on the optimization of multiple trajectories' pose and propose a semantic-based GICP two-stage semantic alignment algorithm, which significantly improves the semantic alignment accuracy, and propose a semantic aggregation algorithm based on lane line logic constraints for lightweight vector map construction, which effectively suppresses the impact of abnormal data on semantic element instantiation. Compared with the state-of-the-art (Qin et al. and Herb et al.), our method reduces the relative accuracy error of fusion mapping by 30%, increases the factor completion rate by about 15%, and improves the mapping efficiency by about 10%. The experimental results demonstrate that our method is highly accurate, lightweight, and robust, and can meet the application accuracy requirements of high-precision mapping. Our method also provides convenience for users to segment and load maps.

**Author Contributions:** Conceptualization, H.S. and B.H.; methodology, H.S.; software, H.S. and Q.H.; validation, H.S. and Y.Z.; investigation, H.S. and J.S.; writing—original draft preparation, H.S.; writing—review and editing, H.S. and J.S. All authors have read and agreed to the published version of the manuscript.

**Funding:** This work was funded by the Ministry of Industry and Information Technology of Things special fund for development funding and the city cooperative guidance project of Guangzhou (No. 2014Y2-00218).

**Data Availability Statement:** No applicable.

**Conflicts of Interest:** The authors declare no conflict of interest.

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
