# Peer review of "A Lightweight High Definition Mapping Method Based on Multi-Source Data Fusion Perception"

_applsci, doi:10.3390/app13053264_

Round 1
Reviewer 1 Report
The manuscript proposes a Lightweight High Definition Mapping Method based on multi-source data fusion perception for multi-trajectory pose optimization using improved two-stage semantic alignment algorithm. Experimental validation is presented with permissible error limits.
Give the full form of abbreviations at their first appearance.
Figure 1, give the YES/No at the decision box, i.e., what will be done, if data is not enough?
There is some dysconnectivity in the statement “….pose of the vehicle can be The absolute pose of the vehicle can be …”, Please correct.
What is the influence of natural/environmental conditions on the colours and brightness of Realtime objects?
The assumptions (for example stated in 3.2.1, 3.2.2 followed in the framework can be listed as a separate section.
Clarity is missing in Figure 4 .
The images need appropriate coordinates/axes in Figure 6
The results presented in table 2 and table 3 are attractive, but a comparative assessment with the prior art is required.
Author Response
We would like to thank you for your careful reading, helpful comments, and constructive suggestions, which has significantly improved the presentation of our manuscript.
We have carefully considered all comments and revised our manuscript accordingly. The manuscript has also been double-checked, and the typos and grammar errors we found have been corrected. In the following section, we summarize our responses to each comment. We believe that our responses have well addressed all concerns from the reviewer. We hope our revised manuscript can be accepted for publication.
Response 1: We thank the reviewer for reading our paper carefully and giving the above positive comments.
Point 2: Give the full form of abbreviations at their first appearance.
Response 2: Thank you for the above suggestion. We have given the full form of abbreviations at their first appearance.
Point 3: Figure 1, give the YES/No at the decision box, i.e., what will be done, if data is not enough?
Response 3: Thank you for pointing out this problem in our manuscript. According to the revised content, we have redrawn Figure 1 adding the YES/No at the decision box.
Point 4: There is some dysconnectivity in the statement “….pose of the vehicle can be The absolute pose of the vehicle can be …”, Please correct.
Response 4: We are very sorry for the mistakes in this manuscript and inconvenience they caused in your reading. We have corrected the typos we found in our revised manuscript.
Point 5: What is the influence of natural/environmental conditions on the colours and brightness of Realtime objects?
Response 5: Thank you for your question. We have conducted multiple experiments in urban natural environments. This paper mainly focuses on the detection and extraction of road feature images in the figure, and does not specifically address the impact of the environment on the experiments. Due to the use of multiple sensors and trajectories for fusion processing, the experimental results have good robustness to natural environments and conditions. Of course, your question is very inspiring to us, and we plan to conduct some targeted experiments in the future.
Point 6: The assumptions (for example stated in 3.2.1, 3.2.2 followed in the framework can be listed as a separate section.
Response 6:Thank you for the above suggestion. In section 3.2,we have listed the assumptions as a separate section.
Point 7: Clarity is missing in Figure 4 .
Response 7:Thank you for pointing out this problem in our manuscript. We have replaced a clearer picture in Figure 4.
Point 8: The images need appropriate coordinates/axes in Figure 6.
Response 8:We are appreciative of the above suggestion. Indeed, the picture in Figure 6 are the semantic map of lane lines obtained after a series of processing of Local Segment Maps on the vehicle, which is the result of the same data in the same section using different methods, so no coordinate axis is added. In addition, because the distance is too short relative to geodetic coordinates, the coordinate scale is difficult to reflect.
Point 9: The results presented in table 2 and table 3 are attractive, but a comparative assessment with the prior art is required.
Response 9:Thank you for your approval and suggestion. In Table 2 and Table 3, we have compared our method with the existing ICP and SICP algotirhm respectively, and compared with our previous improved method as SGICP.

Reviewer 2 Report
In my opinion, this paper is suitable for publication in "applied sciences" for the reasons mentioned below.
1- Paper contributions. The paper's contributions to the general field of mapping method are sound enough to justify publication in such a prestigious journal.
2- Experimental results. The theoretical results obtained in the manuscript are backed up by experimental test. In fact, this is a strong point of the manuscript that claims to be devoted to the multi-source data fusion perception.
3- Bibliography. The literature review presented in the paper is complete. However, it should include more relevant papers.
My only concern is about the conclusion. Indeed, the conclusion is inadequate and does not highlight anything that is specific, important and what happens if it is not used. A lack of constructive arguments for why this paper provides new information is lacking. I suggest that the authors discuss their key objectives, and demonstrate how it is possible to capture some of these intricate details for the purpose of mapping method.
Author Response
We would like to thank you for your careful reading, helpful comments, and constructive suggestions, which has significantly improved the presentation of our manuscript.
We have carefully considered all comments and revised our manuscript accordingly. The manuscript has also been double-checked, and the typos and grammar errors we found have been corrected. In the following section, we summarize our responses to each comment. We believe that our responses have well addressed all concerns from the reviewer. We hope our revised manuscript can be accepted for publication.
Point 1: In my opinion, this paper is suitable for publication in "applied sciences" for the reasons mentioned below.
Response 1: We thank the reviewer for reading our paper carefully and giving the above positive comments.
Point 2: Experimental results. The theoretical results obtained in the manuscript are backed up by experimental test. In fact, this is a strong point of the manuscript that claims to be devoted to the multi-source data fusion perception.
Response 2: Thank you for the above comment. Indeed, as you pointed out, we have done detailed experiments in actual scenarios, and the results have demonstrated our proposed improvement methods is highly accurate, lightweight, and robust, and can meet the application accuracy requirements of high-precision mapping.
Point 3: Bibliography. The literature review presented in the paper is complete. However, it should include more relevant papers.
Response 3: Thank you for pointing out this problem in our manuscript. According to the revised content, we have added more refenren relevant papers in each section.
Point 4: My only concern is about the conclusion. Indeed, the conclusion is inadequate and does not highlight anything that is specific, important and what happens if it is not used. A lack of constructive arguments for why this paper provides new information is lacking. I suggest that the authors discuss their key objectives, and demonstrate how it is possible to capture some of these intricate details for the purpose of mapping method.
Response 4: Thank you for pointing out this problem in our manuscript. We have provided more detailed descriptions of the conclusion in our revised manuscript including the description of the methods we proposed for different stages of HD mapping, and the performance comparison between our methods and state of art in actual experiments.
